# Comprehensive Assessment of Subtyping Methods for Improved Surveillance of Foodborne *Salmonella*

Hang Pan,[a,b] Chenghao Jia,[c] Narayan Paudyal,[a*] Fang Li,[a] Junyong Mao,[c] Xi Liu,[a§] Chenghang Dong,[a] Kun Zhou,[a] Xiayi Liao,[a◇] Jiansen Gong,[d] Weihuan Fang,[a,b] Xiaoliang Li,[a,b] Corinna Kehrenberg,[e] Min Yue[a,b,c,f]

[a]Institute of Preventive Veterinary Sciences & Department of Veterinary Medicine, Zhejiang University College of Animal Sciences, Hangzhou, China

[b]Zhejiang Provincial Key Laboratory of Preventive Veterinary Medicine, Hangzhou, China

[c]Hainan Institute of Zhejiang University, Sanya, China

[d]Jiangsu Institute of Poultry Science, Yangzhou, Jiangsu, China

[e]Institute for Veterinary Food Science, Justus Liebig University Giessen, Giessen, Germany

[f]State Key Laboratory for Diagnosis and Treatment of Infectious Diseases, National Clinical Research Center for Infectious Diseases, National Medical Center for Infectious Diseases, The First Affiliated Hospital, College of Medicine, Zhejiang University, Hangzhou, China

Hang Pan and Chenghao Jia contributed equally to this work. Author order was determined by the responsibility in management and coordination for the research activity planning and execution.

**ABSTRACT** High-resolution and efficient typing for the bacterial pathogen is essential for tracking the sources, detecting or diagnosing variants, and conducting a risk assessment. However, a systematic in-field investigation of *Salmonella* along the food chain has not been documented. This study assessed 12 typing methods, such as antimicrobial-resistance (AMR) gene profile typing, Core Genome Multilocus Sequence Typing (cgMLST), and CRISPR multi-virulence locus sequence typing (CRISPR-MVLST), to evaluate their effectiveness for use in routine monitoring of foodborne *Salmonella* transmission along the poultry production chain. During 2015-16, a total of 1,064 samples were collected from poultry production chain, starting from breeding farms and slaughterhouses to the markets of Zhejiang province in China. A total of 61 consecutive unique *Salmonella* isolates recovered from these samples were selected for genome sequencing and further comparative typing analysis. Traditional typing methods, including serotyping, AMR phenotype-based typing, as well as modern genotyping approaches, were evaluated and compared by their discrimination index (DI). The results showed that the serotyping method identified nine serovars. The gold standard cgMLST method indicated only 18 different types (DI = 0.8541), while the CRISPR-MVLST method detected 30 types (DI = 0.9628), with a higher DI than all examined medium-resolution WGS-based genotyping methods. We demonstrate that the CRISPR-MVLST might be used as a tool with high discriminatory power, comparable ease of use, ability of tracking the source of *Salmonella* strains along the food chain and indication of genetic features especially virulence genes. The available methods with different purposes and laboratory expertise were also illustrated to assist in rational implementation.

**IMPORTANCE** In public health field, high-resolution and efficient typing of the bacterial pathogen is essential, considering source-tracking and risk assessment are fundamental issues. Currently, there are no recommendations for applying molecular characterization methods for *Salmonella* along the food chain, and a systematic in-field investigation comparing subtyping methods in the context of routine surveillance was partially addressed. Using 1,064 samples along a poultry production chain with a considerable level of *Salmonella* contamination, we collected representative isolates for genome sequencing and comparative analysis by using 12 typing techniques, particularly with whole-genome sequence (WGS) based methods and a recently invented CRISPR multi-virulence locus sequence typing (CRISPR-MVLST) method. CRISPR-MVLST is identified as a tool with higher discriminatory power compared with medium-resolution WGS-based typing

Address correspondence to Min Yue, myue@zju.edu.cn.

*Present address: Narayan Paudyal, National Animal Health Research Centre (NAHRC), Nepal Agricultural Research Council (NARC), Lalitpur, Nepal.

§Present address: Xi Liu, Department of Microbiology Laboratory, Shanghai Municipal Center for Disease Control and Prevention, Shanghai, China.

◇Present address: Xiayi Liao, Department of Biology, Indiana University-Purdue University Indianapolis, Indianapolis, Indiana, USA.

The authors declare no conflict of interest.

methods, comparable ease of use and proven ability of tracking *Salmonella* isolates. Besides, we also offer recommendations for rational choice of subtyping methods to assist in better implementation schemes.

**KEYWORDS** *Salmonella*, typing method, CRISPR-MVLST, CoreSNP, cgMLST, antimicrobial resistance, poultry production chain

Foodborne diseases, caused by *Salmonella* and many other pathogens, are critical and sustaining threats to global public health (1, 2). Improved control of foodborne bacterial transmission requires various aspects of investment, such as investigation of epidemiological prevalence, detection of contaminated point, and bacterial typing. The capabilities for quick, reliable, and convenient differentiation of typing approaches are invaluable for diagnosis, treatment and epidemiological surveillance of bacterial infections (3–7).

The conventional typing methods, i.e., bacteriophage typing, serotyping, and biochemical typing, have played important roles in understanding the nature of diversity among clinically relevant bacterial agents (8, 9). Alongside this, antibiogram typing or antimicrobial resistance profiling has been used for epidemiological source prediction or typing purposes (8–11). These phenotyping assays aim to elucidate regional- and national-scale outbreaks due to specific bacterial strains. Though they are also useful for particular purposes, they have several practical limitations that render them unsuitable for comprehensive studies of bacterial population structure or dynamic variants. Nowadays, advanced molecular typing or genotyping have been widely adopted. While having different levels of resolution and time-output efficiency, they require a range of varying expertise for practical implementation.

There are various molecular typing or genotyping methods used in the veterinary public health and food safety field (12). These include: (i) Pulsed-field Gel Electrophoresis (PFGE), (ii) Multi Locus VNTR Analysis (MLVA), (iii) Restriction Fragment Length Polymorphism (RFLP), (iv) Clustered Regularly Interspaced Short Palindromic Repeat (CRISPR), (v) Multilocus Sequence Typing (MLST) and (vi) Whole-Genome Single-Nucleotide Polymorphism (SNP). Currently, there are no recommendations for applying molecular characterization methods for *Salmonella*, although the food industry regularly uses banding pattern-based and sequence-based subtyping methods for incident investigations (13). Two decades ago, US CDC introduced PFGE for routine use in surveillance and set up a PulseNet International network. Its disadvantages, however, were labor-intensity, low robustness, poor comparability of results among different laboratories, and limited resolution in source tracking of disease outbreaks associated with foodborne bacteria, including *Salmonella*. Recently, the whole-genome sequence (WGS) approach started to take place. Although it offers apparent advantages, expensive WGS infrastructure and use of downstream bioinformatic toolkits remain key bottlenecks for academic and surveillance staff.

In general, phenotypic methods are not promising for tracking sources, as in most cases, they are very time- and labor-intense, and usually require well-trained technicians. Nevertheless, serotyping and AMR profiling are still routinely used in food safety, particularly for typical foodborne pathogens, i.e., *Salmonella*. Given the increasing need to detect emerging clones or hazards, identifying the distinct types and pinpointing the source of *Salmonella* isolates is critical for improving surveillance and implementing control measures for such risks along the production chain (14, 15). Although a range of typing methods have been presented, to our knowledge, no studies have systematically examined or compared different typing methods in the field for their practical application potential.

CRISPR multi-virulence locus sequence typing (CRISPR-MVLST) was proved to have good discriminatory power, but only a few studies reported other traits of this method. Previous research showed that bacteria from distant geographic locations had extremely different spacer arrangements because of the existence of unique phage or plasmid pools in those different geographic locations (16). It is suggested that spacer

**TABLE 1** Samples and prevalence rate

| Sampling place | Sampling size | Source | Positive samples | Prevalence |
|---|---|---|---|---|
| Huzhou | 40 | Breeding farm A | 3 | 7.50% |
| Hangzhou | 42 | Breeding farm B | 6 | 14.29% |
| Yiwu | 64 | Breeding farm C | 6 | 9.37% |
| Yiwu | 46 | Slaughterhouse A | 13 | 28.26% |
| Yiwu | 448 | Slaughterhouse B | 92 | 20.54% |
| Huzhou | 259 | Slaughterhouse C | 47 | 18.15% |
| Hangzhou | 42 | Supermarket | 30 | 71.43% |
| Hangzhou | 22 | Market A | 5 | 22.73% |
| Hangzhou | 47 | Market B | 9 | 19.15% |
| Hangzhou | 54 | Market C | 42 | 77.78% |
| Total | 1,064 | | 253 | 23.78% |

arrangements may be a good indicator of bacterial adaptation to diversified microenvironments (17). Nevertheless, CRISPRs may evolve much faster than virulence genes (18). Besides, the loss or duplication of a single spacer and its associated direct repeat could frequently cause small allelic differences in CRISPR arrays between different *Salmonella* isolates (19, 20). While CRISPR-MVLST sequence type (CST) was reported to be associated with AMR in *Salmonella* Typhimurium (21), as far as we know, few studies have reported its relationship with the AMR gene and compared it with other genetic features.

For proof of concept, we used the newly produced data in the poultry production chain with a focus on *Salmonella*. We aimed to provide a reference for the practical application of 12 methods in the context of routine epidemiological surveillance, including serotyping, MLST, MIC profile typing, AMR profile typing, CRISPR typing (CT), CRISPR-MVLST, Virulence Factor (VF) gene profile typing, plasmid profile typing, AMR gene profile typing, core genome multilocus sequence typing (cgMLST), whole-genome MLST (wgMLST), and CoreSNP typing. Our results showed that CRISPR-MVLST would be the best choice for traceability and ease of use for *Salmonella* isolates along the food chain. Further, we proved close correlations between CRISPR-MVLST results and AMR genes or VF genes.

## RESULTS

**Prevalence and distribution of *Salmonella*.** Among the 1,064 samples collected along the poultry production chain, a total of 253 *Salmonella* positive samples were detected, representing an overall prevalence rate of 23.78% (Table 1). The prevalence rates of *Salmonella* in breeding farms, slaughterhouses, and markets were 10.27% (15/146), 20.19% (152/753), and 52.12% (86/165), respectively. Our results showed a high level of contamination along the poultry production chain, especially in the market, which could be a potential risk for consumers. No isolate was found in the samples collected from breeding farms, possibly because positive samples were few, and most of the contaminations were caused by *Salmonella* Gallinarum biovar Pullorum, which are highly avian-adapted and grew too slow to be isolated in selective enrichment medium used during the sampling periods (22). Due to dominance of *S*. Pullorum in China (23, 24) and its host restriction, any contamination in breeding farms is of lesser priority within the scope of the foodborne pathogen. Therefore, this study focused on the contamination in slaughterhouses and markets. We eliminated the copy isolates within the same sample origin (copy isolates here are defined as clones with the same colony morphology during isolation, serovar and ST from the same sample) and selected a representative unique collection of 61 *Salmonella* isolates, including 40 isolates from slaughterhouses and 21 isolates from markets, to further evaluate subtyping methods.

**Phenotyping analysis: AMR phenotype-based typing and serotyping.** Phenotypic AMR of the 61 isolates was evaluated using the MIC of 15 antimicrobials, and the results are summarized in Fig. S1 and Table S1. When the isolates categorized as intermediate were also considered as resistant, all the studied isolates (*n* = 61) were resistant to more than 3 antimicrobial classes and were classified as multi-drug resistance

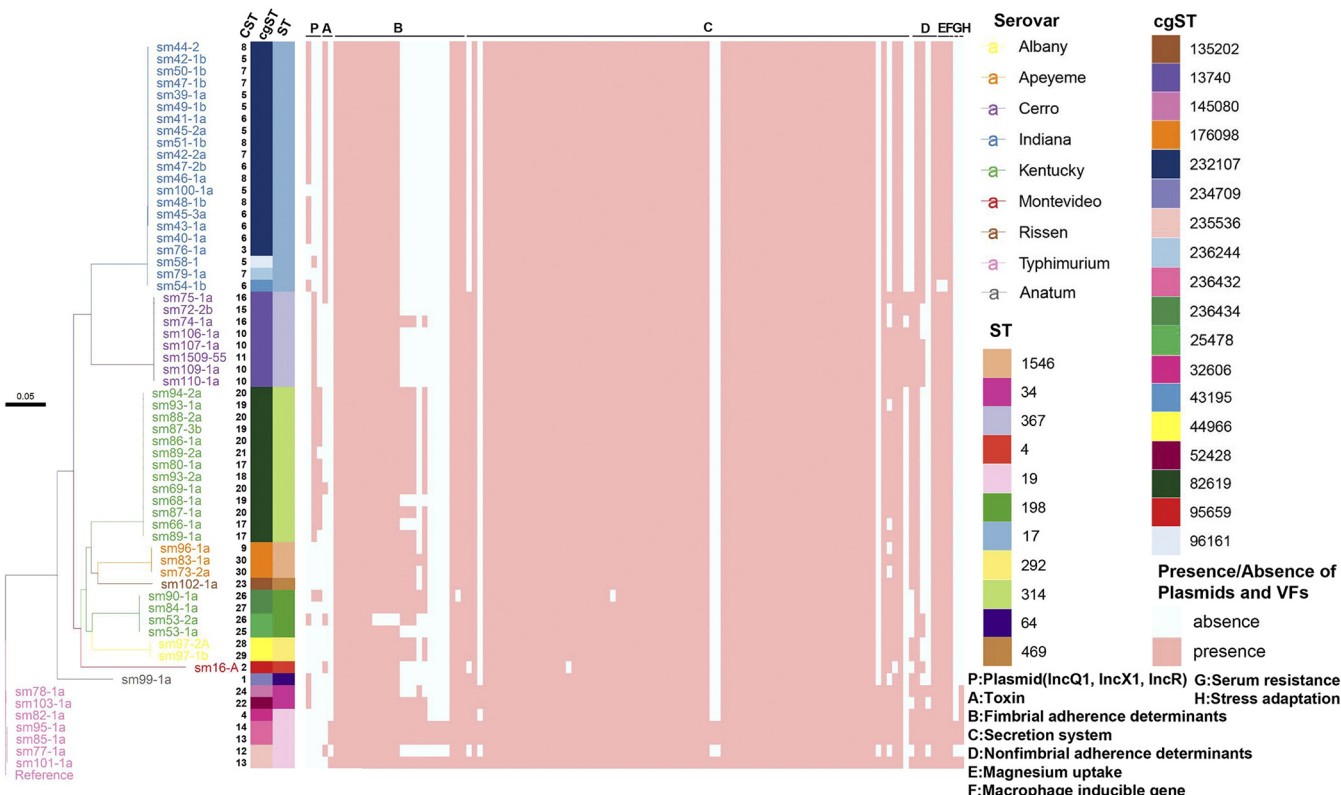

**FIG 1** Results of *Salmonella* typing and virulence gene detection of 61 isolates. A phylogenetic tree was built to show the genetic relationship of 61 isolates and the visualization of core genome SNP typing. Different serovars were labeled with different colors, as shown in the right part near the heatmap. A parallel matching heatmap was aligned to the phylogenetic tree. The left side of the heatmap shows the CST, cgST, and ST results. Different colors are used to distinguish different types, as noted on the right side of the heatmap. The right part shows a detailed matrix of plasmids (IncQ1, IncX1, IncR), and VF genes. The presence of a plasmid or VF gene in an isolate is marked as pink, otherwise, it is marked as light blue.

(MDR) (Fig. S2). Additionally, we found the resistance rate varied dramatically among 15 examined antimicrobials (Fig. S1). Therefore, it would be appropriate to convey 2 AMR phenotype-based typing methods here. All isolates were categorized using the MIC value profile (MIC data matrix of 15 antimicrobials), with 51 types identified using the AMR profile (AMR data matrix of 15 antimicrobials).

The 61 studied isolates comprised 9 serovars (Table S1). Serotyping by the slide agglutination showed identical results to the *in silico* prediction method. One exception resulted from a misjudgement during the slide agglutination test and was revealed by the *in silico* prediction results.

**Non-WGS-based genotyping analysis: MLST, CT, and CRISPR-MVLST.** The 61 studied *Salmonella* isolates comprised 11 STs (Table S1), including *S.* Indiana ST17 (34.43%; 21/61), *S.* Kentucky ST314 (21.31%; 13/61), *S.* Kentucky ST198 (6.56%; 4/61), *S.* Cerro ST367 (13.11%; 8/61), *S.* Typhimurium ST19 (8.20%; 5/61), *S.* Typhimurium ST34 (3.28%; 2/61), *S.* Apeyeme ST1546 (4.92%; 3/61), *S.* Albany ST292 (3.28%; 2/61), *S.* Anatum ST64 (1.64%; 1/61), *S.* Montevideo ST4 (1.64%; 1/61), and *S.* Rissen ST469 (1.64%; 1/61). The MLST results obtained from the Sanger sequences were consistent with those from WGS. Among all 61 isolates, MLST discriminated 11 types while serotyping discriminated 9 types. The outcomes of these 2 methods are shown with a similar color scheme and appear well-matched (Fig. 1).

The typing and cluster analysis of isolates was undertaken by determining CRISPR-pattern-based on the spacer sequences of CRISPR1 (C1) and CRISPR2 (C2) loci. The phylogenetic tree and heatmap were generated only for serovars with a high number of isolates, including Indiana, Typhimurium, Cerro, and Kentucky, in which 2, 6, 2, and 2 CRISPR-patterns were identified, respectively (Fig. 2). The spacer sequences contained with different CRISPR-patterns are given in detail in Table S2. The CRISPR-MVLST

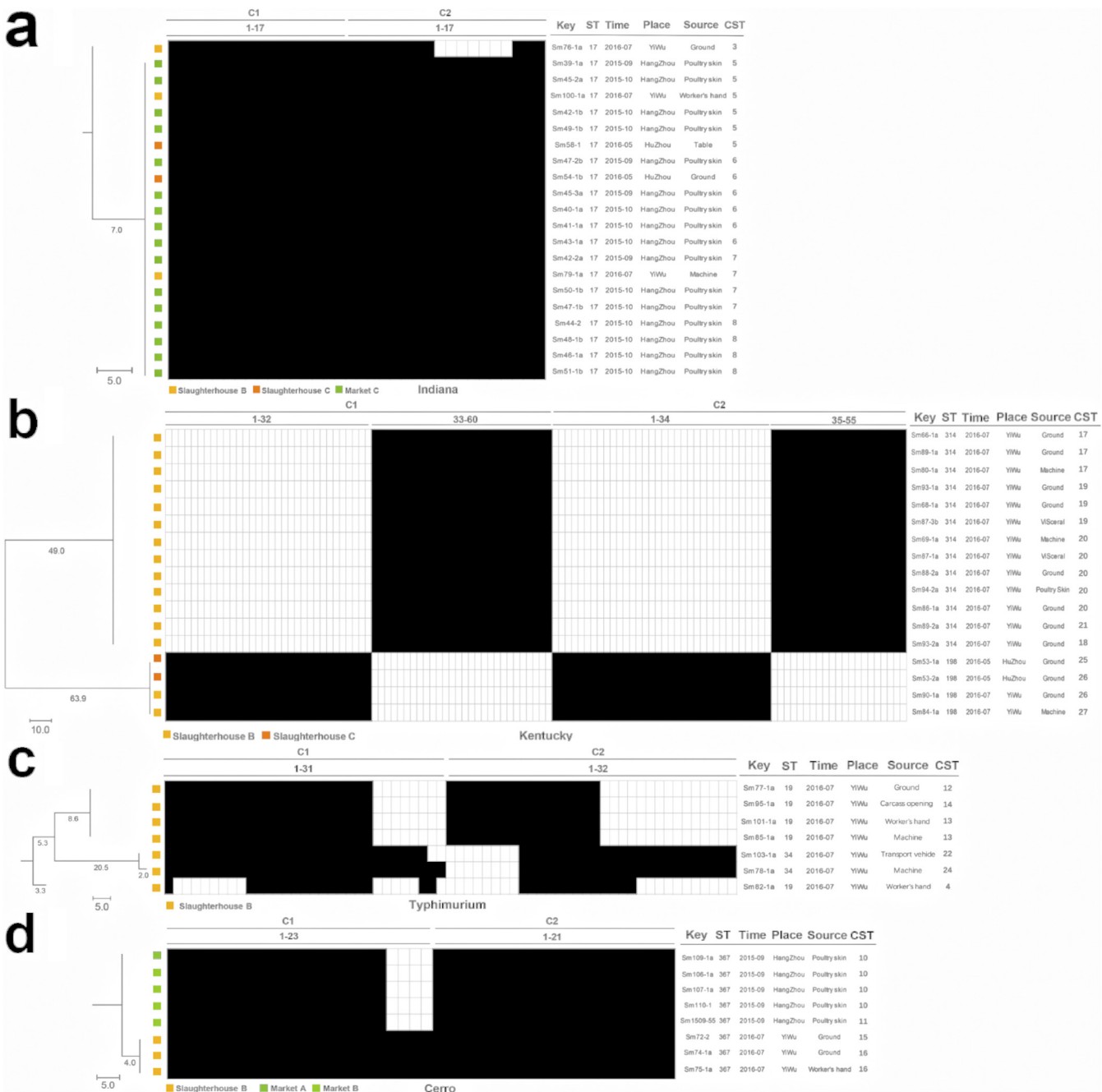

**FIG 2** CRISPR-pattern of the studied *Salmonella* serovars, including Indiana (a), Kentucky (b), Typhimurium (c), and Cerro (d). CRISPR cluster analysis diagram of 4 serovars: (a) Indiana, (b) Typhimurium, (c) Cerro, (d) Kentucky. These phylogenies were made to cluster isolates based on their CRISPR spacer profiles (CRISPR-patterns). Key variables and isolate information are marked around the trees and heatmap of the CRISPR-pattern matrixes: CRISPR 1 is abbreviated as C1 and CRISPR 2 as C2. The tree scale bar represents a standard distance estimated by a neighbor-joining method. Time of sample collection (time), location of sample isolation (place), CRISPR-MVLST sequence type (CST), MLST sequence type (ST). The yellow and orange color represents the isolate from slaughterhouses A and B, respectively. The dark green, light green and standard green color represent the isolate from market A, B and C, respectively. In the CRISPR-pattern heatmap, each square represents a spacer sequence and those marked in black indicates the presence of the spacer in that isolate, while remaining blank indicates the absence of the spacer. Due to the excessive number of spacers, they are marked and labeled with numbers on the diagram for convenience. Take (b) as an example, 1-32 means spacers of C1 from the 1st to the 32nd; 1-55 means all the spacers of C2 from the first to the end. A line is drawn to segregate the necessary section and facilitate locating the spacer.

sequence types (CSTs) were classified by adding the loci information of 2 VF genes *sseL* and *fimH* to 2 basic CRISPR loci as previous studies reported (19, 25–27). Our findings showed the presence of 30 different CSTs among these 61 *Salmonella* isolates (Table S3). In serovar Indiana, 17 types of C1 spacer and 17 types of C2 spacer were detected, with 1 ST (ST17) and 5 CSTs (Fig. 2a). In serovar Kentucky, 60 types of C1 spacer and 55

**TABLE 2** Comparison of three examined phenotyping methods in this study[a]

| Assessment indicator | Serotyping | AMR profile | MIC profile |
|---|---|---|---|
| Repeatability | Good | Moderate | Moderate |
| Reproducibility | Moderate[b] | Moderate | Poor |
| Discriminatory power (DI)[c] | 0.7820 | 0.9940 | 1 |
| Discriminated types | 9 | 51 | 61 |
| Scheme standardized or not[d] | Yes | No | No |
| Ease of interpretation of data generated[e] | Good | Excellent | Excellent |
| Ease of use | Poor to moderate[f] | Good to moderate[g] | Poor to moderate[h] |
| Throughput | No | Yes | Yes |
| Cost | Moderate to high[i] | Moderate to high[j] | High |
| Time required (days)[k] | 2~17# (usually > 5 days for expt) (13) | 3 | 3 |

[a]Ranking: 1. DI in order (good to poor): MIC profile = wgMLST > CoreSNP > AMR profile > CRISPR-MVLST > VF gene profile > AMR gene profile > plasmid profile > cgMLST > CRISPR > MLST > serotyping. 2. Ease of use (good to poor): CRISPR > CRISPR-MVLST > MLST > cgMLST = wgMLST > plasmid profile = AMR gene profile = VF gene profile = CoreSNP > AMR profile > MIC profile > serotyping. 3. Cost (<u>low to high</u>): CRISPR < CRISPR-MVLST < MLST = VF gene profile = AMR gene profile = plasmid profile < wgMLST = cgMLST = coreSNP < AMR profile < serotyping < MIC profile. 4. Time required (<u>short to long</u>): MLST < CRISPR < CRISPR-MVLST < AMR profile = MIC profile < plasmid profile = VF gene profile = AMR gene profile < wgMLST < CoreSNP = cgMLST < serotyping.
[b]We summarized three phenotyping methods in a similar manner of a previous study (28).
[c]Using the Discriminatory index (DI) for a description of discriminatory power.
[d]If there is 1~2 universally acknowledged standard for this typing or not.
[e]Intended as unequivocal interpretation.
[f]Using *in silico* prediction will be easier and faster.
[g]Using disc agar diffusion test will be easier.
[h]Become easier if the lab is doing antimicrobial resistance-related research.
[i]Using *in silico* prediction will be cheaper.
[j]Using disc agar diffusion test will be more affordable.
[k]The approximate number of days to get typing results is estimated by excluding the interval of time to obtain a single pure colony suitable to be handled by the method.

types of C2 spacer were detected, with 2 STs (ST314 and ST198, each ST showed a fully distinguished CRISPR-pattern) and 8 CSTs (Fig. 2b). In serovar Typhimurium, 31 types of C1 spacer and 32 types of C2 spacer were detected, with 2 STs (ST19 and ST34) and 6 CSTs (Fig. 2c). In serovar Cerro, 23 types of C1 spacer and 21 types of C2 spacer were detected, with 1 ST (ST367) and 4 CSTs (Fig. 2d).

**WGS-based genotyping analysis: cgMLST, wgMLST, CoreSNP, AMR gene, VF gene, and plasmid profile typing.** The whole-genome sequences of these *Salmonella* isolates (*n* = 61) were analyzed to predict the plasmid replicons, AMR genes, VF genes, and calculate cgST, CoreSNP type, and wgMLST type (summarized in Table S1, wgMLST profile matrix in Table S4).

The results of plasmid replicon prediction showed existence of 18 different plasmids. All isolates were divided into 18 types using plasmid profiles (data matrix of plasmid presence/absence table).

The results of AMR gene detection showed the presence of 59 different AMR genes and 2 patterns of AMR chromosomal mutations at varying levels (Fig. S3 and Fig. S4). These isolates were divided into 32 types using the AMR gene profile (data matrix of AMR gene presence/absence).

The results of VF gene detection showed the presence of 156 genes (117~139 gene per isolate, Fig. 1, and Table S1). The isolates were divided into 36 types using VF gene profile (data matrix of VF gene presence/absence).

**Comparative analysis of various typing methods.** The comparison of discriminatory power among the 6 non-WGS genotyping or phenotyping methods in this study, including serotyping, MLST, CRISPR-MVLST, CRISPR, AMR profile, and MIC profile, were carried out based on the discriminatory index (DI). The results showed that, except for AMR phenotype-based methods, the CRISPR-MVLST method provided higher discrimination power of the 61 studied isolates (DI = 0.9628), followed by the CT (DI = 0.8377), MLST (DI = 0.8158), and serotyping method (DI = 0.7820) (Table 2 and 3).

On the other hand, for 6 WGS-based genotyping methods, cgMLST discriminated 18 different types among these 61 isolates and thus presented a DI of 0.8541, which was higher than that of 7-gene legacy MLST. However, our results showed that cgMLST could not distinguish isolates with close relationships, i.e., *S.* Cerro and *S.* Kentucky, while CRISPR-MVLST could clearly distinguish these isolates. The cgMLST

**TABLE 3** Comparison of nine examined genotyping methods in this study[a]

| Assessment indicator | non-WGS-based typing | | | WGS-based typing | | | | | |
| | Low-resolution[b] | | High-resolution | Medium-resolution | | | | High-resolution | |
| | 7-gene legacy MLST | CRISPR | CRISPR-MVLST | cgMLST | Plasmid profile | AMR gene profile | VF gene profile | CoreSNP | wgMLST |
|---|---|---|---|---|---|---|---|---|---|
| Repeatability | Excellent | Excellent | Excellent | Excellent | Excellent | Excellent | Excellent | Excellent | Excellent |
| Reproducibility | Excellent | Excellent | Excellent[c] | Excellent | Excellent | Excellent | Excellent | Excellent | Excellent |
| Discriminatory power (DI)[d] | 0.8158 | 0.8377 | 0.9628 | 0.8541 | 0.8852 | 0.9361 | 0.9497 | 0.9967 | 1 |
| Discriminatory types | 11 | 15 | 30 | 18 | 18 | 32 | 36 | 56 | 61 |
| Scheme standardized or not[e] | Yes | Yes | Yes | Yes | No | No | No | Not yet | Not yet |
| Ease of interpretation of data generated[f] | Excellent | Excellent | Excellent | Good | Excellent | Excellent | Excellent | Moderate | Moderate |
| Ease of use | Good to moderate[g] | Moderate | Moderate | Moderate | Good | Good | Good | Poor | Moderate |
| High throughput | Yes | Yes | Yes | Yes | Yes | Yes | Yes | Yes | Yes |
| Cost | Moderate | Low | Low | Moderate | Moderate | Moderate | Moderate | Moderate | Moderate |
| Time required (days)[h] | 1~2 | 1~2 | 1~2 | 2~7 | 2~7 | 2~7 | 2~7 | 2~7 | 2~7 |

[a]Rankings 1. DI in order (good to poor): MIC profile = wgMLST > CoreSNP > AMR profile > CRISPR-MVLST > VF gene profile > AMR gene profile > plasmid profile > cgMLST > CRISPR > MLST > serotyping. 2. Easy of use (good to poor): CRISPR > CRISPR-MVLST > MLST > cgMLST = wgMLST > plasmid profile = AMR gene profile = VF gene profile > CoreSNP > AMR profile = MIC profile = serotyping. 3. Cost (low to high): CRISPR < CRISPR-MVLST < MLST = VF gene profile = AMR gene profile = plasmid profile < wgMLST = cgMLST = coreSNP < AMR profile < serotyping < MIC profile. 4. Time required (short to long): MLST < CRISPR < CRISPR-MVLST < AMR profile = MIC profile < plasmid profile < VF gene profile = AMR gene profile < wgMLST < CoreSNP = cgMLST < serotyping.
[b]For reading ease, Low/Medium/High resolution classification is set up for genotyping methods according to their DI results in this study.
[c]We summarized nine methods in a similar manner of a previous study (28).
[d]Using the Discriminatory index (DI) for a description of discriminatory power.
[e]If there is 1~2 universally acknowledged standard for this typing or not.
[f]Intended as unequivocal interpretation.
[g]Using *in silico* prediction will be easier.
[h]The approximate number of days to get typing results is estimated by excluding the interval of time to obtain a single pure colony suitable to be handled by the method.

showed a lower DI than most of the other tested methods, including the non-WGS-based method CRISPR-MVLST. VF gene profile typing (DI = 0.9497), AMR gene profile typing (DI = 0.9361) and plasmid profile typing (DI = 0.8852) gave a moderate level of discrimination power. Additionally, our results showed that WGS-based high-resolution methods presented remarkable discrimination power (DI > 0.9960, [Table 3]).

Thus, based on these findings, the typing methods can be arranged in decreasing order of their DIs as, MIC profile ≈ wgMLST > CoreSNP > AMR profile > CRISPR-MVLST > VF gene profile > AMR gene profile > plasmid profile > cgMLST > CRISPR > 7-gene legacy MLST > Serotyping.

A summary table of molecular typing methods, including MLST, was reported previously (28). We summarized 12 tested methods in a similar manner (Table 2 and 3) to provide a comprehensive evaluation, along with a ranking heatmap of 12 methods (Fig. 3).

**CRISPR-MVLST tracking *Salmonella* isolates in the poultry production chain.** After demonstrating advantageous traits of the CRISPR-MVLST method, we tried to find out if the method could be applied to *Salmonella* source-tracking. The available results not only accurately revealed major lineages (STs) of 61 studied *Salmonella* isolates, but also clearly suggested cross-contamination points in 4 different serovars.

In serovar Indiana (Fig. 2a), all isolates recovered from markets came from the market C in Hangzhou. Isolate Sm100-1a (from the hand of a worker in slaughterhouse B in Yiwu), isolates Sm45-2a and Sm39-1a (from the skin of chilled poultry carcasses in market C) clustered together with consistent CRISPR-patterns and CSTs. Similar results were obtained in CST 6 and 7, including Sm54-1b (from slaughterhouse C in Huzhou) and Sm47-2b from market C. In addition, there was only one locus difference (in C2) between Sm76-1a (CST 3) and Sm39-1a (CST 5). This supported the close genetic relationship among isolates from market C and those from slaughterhouse B.

In serovar Kentucky (Fig. 2b), isolates Sm69-1, Sm87-1a, Sm88-2a, Sm94-2a, and Sm86-1a (recovered from equipment in the lairage area, viscera room, and skin before

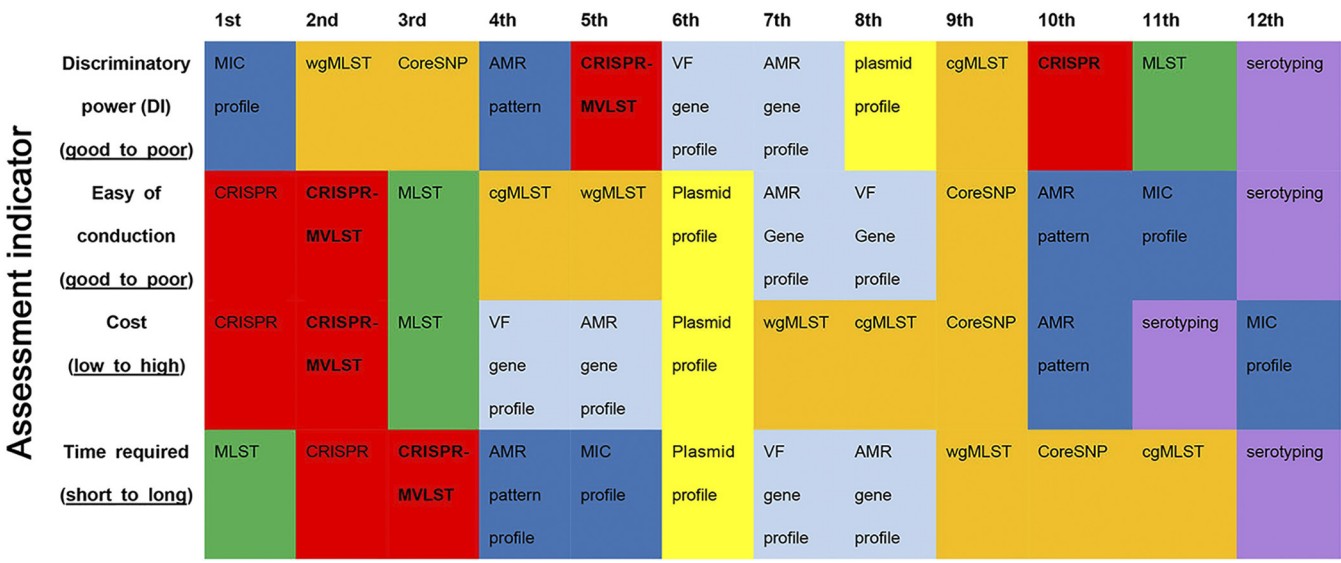

**FIG 3** A visualized ranking of 12 examined typing methods in this study. The *x* axis shows the ranking position for 12 methods. The *y* axis shows all assessment indicators for these methods. Blocks in red represent methods using spacer sequences of CRISPR loci. Blocks in orange represent methods using a relatively global loci/site at a whole-genome level. Blocks in green represent the 7-gene legacy MLST method. Blocks in deep blue represent methods based on phenotypic antimicrobial resistance (AMR) data. Blocks in light blue represent methods based on AMR or VF loci. Blocks in light blue represent methods using plasmid profiles. Blocks in purple represent methods based on the White–Kauffman serotyping scheme.

washing in one slaughterhouse) showed the same CST, suggesting that there may have been a strong viscera contamination before leaching. *S.* Kentucky probably had also contaminated the ground of the lairage, the equipment of the plucking room, and the ground of the carcass washing room (Sm68-1: CST19, Sm80-1a: CST17, Sm93-2a: CST18), suggesting that they could survive for a long time on the ground or equipment. The CRISPR-patterns of Sm90-1a (from the storage room) and Sm84-1 (from the viscera room) from slaughterhouse B (Yiwu) were identical to the other 2 isolates (Sm53-1a and Sm53-2a) from the lairage area of slaughterhouse C in Huzhou (Fig. 2b). The CSTs of Sm90-1a and Sm53-2a were also identical. It is likely that slaughterhouses B and C might have been contaminated with clones of the same origin in the breeding farms.

In serovar Typhimurium (Fig. 2c), it was found that ST19 and ST34 were not fully distinguished; Sm82-1a (ST19) shared all its spacers with Sm78-1a (ST34) but only 80% of its spacers with other ST19 isolates. Besides, there were differences in three (75%) loci of CRISPR-MVLST profiles between the 2 ST34 isolates. It was worth noting that the isolate recovered from the hands of workers at the end of the slaughter chain had high genetic congruence to those from the ground of plucking room, the machine of viscera room and carcass opening before washing (Fig. 2c, and Table S1 and S3). For example, the CST of Sm101-1a (from the worker in the packaging room) was the same as that of Sm85-1a (from the machine in the viscera room). This suggested the presence of a certain degree of cross-spatial contamination in slaughterhouse B.

In serovar Cerro (Fig. 2d), a high similarity of CRISPR sequences (sharing 91% spacers, 40/44) was observed between 3 isolates (from slaughterhouse B in Yiwu) and 5 isolates (from market A or market B in Hangzhou), suggesting that slaughterhouse B could be a potential source of *S.* Cerro in market A and B. Additionally, 4 out of these 5 isolates (Sm106-1a, Sm107-1a, Sm110-1, and Sm1509-55), which were all recovered from the skin of chilled poultry in market B, presented the identical CRISPR-pattern, and the former 3 were all identified as CST 10, suggesting the existence of frequent cross-contamination in market B. In the scalding room in slaughterhouse B, 2 isolates sharing one CST (CST 16) were isolated from the worker's hand (Sm75-1a) and the ground (Sm74-1a).

**Correlation analysis between CRISPR-MVLST results and genetic features.** CRISPR-MVLST was further examined to investigated if it could deliver indications of genetic features including AMR determinants (ARD), VF genes, and plasmid replicons from certain perspective. We also compared it with some standard methods as controls, including serotyping, MLST and cgMLST. Firstly, to find genetic features that were locally associated with certain CST in the poultry production chain, such as ARDs or major plasmid replicons, additional heatmaps were projected for visualization (Fig. 1 and Fig. S3).

Secondly, to locate significant correlation between CRISPR-MVLST results and genetic features, we tried to number the CST in the order of the total number of C1 and C2's spacers (from small to big) and calculated the Spearman's rank correlation coefficient between the numerical number of CST (nCST) and ARD/VF gene/plasmid. Some close correlation (correlation coefficient $>0.6$ or $<-0.6$, $P < 0.01$) was detected.

The most abundant AMR genes were *aph*(6)-*Id* (67.21%) encoding resistance to aminoglycosides (Fig. S4), *tet*(*A*) (72.13%) encoding resistance to tetracyclines, *floR* (63.93%) encoding resistance to phenicols, and *sul1* (60.66%) encoding resistance to sulfonamides, while no significant feature was found to be associated with certain CST. Nevertheless, the isolates with the same ST or cgST carried similar AMR gene patterns. nCST was close negative correlation with *aadA5*, *dfrA17*, *oqxA,* and *oqxB* (correlation coefficient was $-0.64$, $-0.68$, $-0.69$, and $-0.69$, respectively, $P < 0.01$).

For VF genes, *pefABCD* gene cluster, *rck*, *gogB*, *spvC*, *spvR*, and *sodCI* were only detected in CST13 and CST14 of *S*. Typhimurium ST19 (Fig. 1). *cdtB* encoding typhoid toxin-producing was closely related to *S*. Indiana ST17 while *spvB* was closely related to *S*. Typhimurium ST19. The *lpf* gene cluster, encoding the long polar fimbriae, mediating attachment to the Peyer's patches, was associated with *S*. Kentucky ST314 and ST198, *S*. Typhimurium ST19, *S*. Apeyeme ST1546, and *S*. Albany ST292. A high correlation between *sspH1* and *S*. Cerro ST367 was detected. *sspH2* was detected in all serovars except *S*. Indiana ST17. Moreover, *sseI/srfH* was highly correlated with ST19 and ST34, while *ratB* (non-fimbrial adhesin) was detected in *S*. Indiana ST17, *S*. Kentucky ST198, *S*. Cerro ST367, *S*. Typhimurium (ST19, ST34), and *S*. Montevideo ST4. nCST was close positive correlation with *avrA*, *lpfA*, *lpfB*, *lpfC*, *lpfE,* and *sseK1* (correlation coefficient was 0.74, 0.67, 0.67, 0.67, 0.64, and 0.63, respectively, $P < 0.01$), and was close negative correlation with *cdtB*, *hsiC1/vipB,* and *ratB* (the correlation coefficient was $-0.64$, $-0.73$, and $-0.62$, respectively, $P < 0.01$).

Of the 18 different plasmids, among which IncX1 (37.70%; 23/61) and IncQ1 (34.43%; 21/61) were the most prevalent in these studied isolates. Col156 was predicted to exist in CST6 only. Moreover, the distribution of 18 plasmids among serovars showed that *S*. Indiana ST17 harbored more diversified plasmids ($n = 9$), followed by *S*. Typhimurium ST19 ($n = 6$) and *S*. Kentucky ST198 ($n = 5$) (Fig. S3). No close correlation was detected between nCST and plasmid replicons.

The results of chromosomal AMR mutation detection showed 2 mutation patterns in the quinolone-resistance-determining region (QRDR), including the pattern with a single mutation in the *gyrA* gene (S83F) (39.3%) and a pattern with double mutations in the *gyrA* gene (S83F and D87G) (6.6%) (Fig. S4). No close correlation was detected between nCST and chromosomal AMR mutations.

These results indicated close correlations between CRISPR-MVLST results (or total No. of C1&C2's spacers) and ARG or VF genes but not between CRISPR-MVLST results (or total No. of C1&C2's spacers) and plasmids or AMR mutations in *Salmonella* isolates. And some VF genes were found to be associated with certain CSTs.

## DISCUSSION

The poultry production chain is considered the main vehicle for Salmonella human infections. Here, we report a high prevalence of *Salmonella* in different stages of production, including breeding farm, slaughterhouse, and market. Additionally, we demonstrate that the contamination rate increased from upstream (the stunning point at

the slaughterhouse) to downstream (storage/sales of finished products at the markets) of the poultry value chain, reflecting the issue of cross-contamination in the slaughter-house, transport, and market as in previous studies (29–31). Moreover, the study also shows that majority of the *Salmonella* isolates were highly MDR, and harbored various ARDs and VF genes, which is considered a significant concern to public health.

Currently, different typing methods have been used to track the movement of *Salmonella* along with the food chain and to correlate the disease outbreaks with the probable source. Serotyping was considered the classical yet innovative typing method for *Salmonella*, which could identify the major groups that caused human salmonellosis (32, 33). Thereafter, several typing methods based on the analysis of amplified, re-stricted, or sequenced DNA profiles of *Salmonella* isolates have been developed to pro-vide more accurate typing results (13, 34–36).

The choice of the appropriate method is influenced by various factors, including (i) the discriminatory ability to distinguish between non-clonal isolates, (ii) the ability to generate interpretable data, (iii) the reproducibility of results among different person-nel and laboratories, (iv) the time required to return typing results, (v) the need for a standardized scheme, and (vi) the technical complexity, including bioinformatics skills and the resources in terms of equipment, personnel and cost (34). A full evaluation would determine the most appropriate method for source-tracking or variant detect-ing along the food chain.

In the last 2 decades, PFGE has been used as the gold standard typing method by the PulseNet network before being emergence of WGS. Despite its advantages, PFGE has some inherent limits: time-consuming and low discrimination power for all unre-lated isolates. MLVA is used as a typing method that can compensate for the low dis-criminatory power of PFGE in some *Salmonella* serovars but probably will be replaced by WGS (13). Moreover, MLST based on the sequence analysis of seven housekeeping genes, or its recent version based on the core genome sequences (cgMLST), has been used to provide appropriate sequence types of *Salmonella* isolates and has succeeded in discriminating some AMR-related *Salmonella* clones like *S.* Kentucky ST198, *S.* Indiana ST17, the monophasic variant of *S.* Typhimurium ST34 (37–39). Furthermore, CRISPR typing and its updated version CRISPR-MVLST have been recently used to provide high discrimination ability of *Salmonella* isolates, especially CRISPR-MVLST (19, 26, 40).

Although WGS is currently used as a gold standard method for typing foodborne patho-gens, it seems that the traditional phenotypic serotyping and Sanger sequencing approaches are more suitable for initial monitoring, especially in developing countries. Our findings showed that among the 61 isolates of our study, serotyping identified 9 serovars, 7-gene leg-acy MLST identified 11 STs, and CRISPR-MVLST identified 30 subtypes, while cgMLST identified 18 cgSTs, VF gene profile identified 36 subtypes, and CoreSNP identified 56 subtypes. Altogether, CRISPR-MVLST (DI = 0.9628) provided a high discrimination power compared with most of other methods, except CoreSNP, wgMLST or 2 AMR phenotype-based methods. CRISPR-MVLST has shown a high discriminatory ability (DI = 0.980) in typing *S.* Dublin recov-ered from humans and animals (41). CT identified 76 types with a discriminatory power of 97.6% among 180 clinical *Salmonella* strains isolated during 2017–2018 (42). Several previous studies have proven the efficiency of CRISPR-MVLST in typing different *Salmonella* serovars, including Typhimurium, Newport, and Enteritidis, suggesting the use of this method to com-plement and validate results obtained by PFGE (43–46). A study has demonstrated that CRISPR-MVLST could separate the common PFGE patterns of *S.* Heidelberg, providing signifi-cantly greater discriminatory power, and proposed using CRISPR-MVLST as an alternative to PFGE (26). Based on the performance shown in this study, CRISPR-MVLST appeared as an ideal typing method with high discriminatory power, proven ability to track isolate and general applicability (intended as standardized, reproducible, and low requirements in technicality level and equipment) for foodborne pathogen surveillance (Table 3). For 2 AMR phenotype-based methods, though with considerable DIs, mediocre repeatability, poor general applicabil-ity, and high cost imply less efficiency during surveillance, which may not be acceptable for most sentinel laboratories. More importantly, the AMR gene and plasmid are frequently linked

with mobile elements, therefore, bacterial strain might rapidly change due to high frequency of horizontal gene transfer.

We hypothesize that most laboratories have at least the ability to perform the basic serotyping by phenotypic method and the best subtyping methods for different levels of laboratory expertise are summarized here:

(i) *Without any sequencing ability, only have phenotyping ability*.

BEST OPTION: Serotyping or (for local surveillance only) AMR phenotype-based methods.

(ii) *Sanger sequencing available but without WGS*.

BEST OPTION: CRISPR–MVLST in general, and AMR phenotype-based methods can be used as supplements if available.

(iii) *Sanger sequencing & WGS available but without good bioinformatics skill or calculation capability*.

BEST OPTION: wgMLST (online website based) or CRISPR–MVLST.

(iv) *Sanger sequencing & WGS available with good bioinformatics skill & calculation capability*.

BEST OPTION: wgMLST or CoreSNP.

Our findings showed a high efficiency of CRISPR-MVLST in tracking the source of *Salmonella* along the poultry production chain. The subtype CST26 was identified in the samples from the waiting room of slaughterhouse C and the storage room of slaughterhouse B, suggesting that the isolates could have come from the same farm, where the animals were colonized with *Salmonella* before being transported to different slaughterhouses. Additionally, the findings demonstrated the presence of isolates such as CST13, CST19, and CST20 with the same CRISPR-pattern, at different processing steps in slaughterhouse B, indicating the persistence of *Salmonella* isolates after applying sanitization operations and a high frequency in contamination as well as cross-contamination of poultry carcass in this slaughterhouse. The slaughtering process is a critical step in the poultry production chain, and bacteria can disseminate from intestinal content during the evisceration process and then contaminate and/or cross-contaminate poultry carcasses, facilities, and workers' hands along with the slaughtering process steps (15). The subtypes CST5 and CST7 were identified both in slaughterhouse B and poultry carcasses in market C, indicating that the poultry might have been contaminated in the slaughterhouse before being supplied to the market. The same reasoning is valid for the CST6 subtype isolates recovered from slaughterhouse C and poultry carcasses sold in the market C. These results also suggest the implementation of robust and efficient disinfection systems and personal hygiene to prevent and control the dissemination of *Salmonella* at those critical points of the poultry production chain. Interestingly, there was no obvious evidence supporting that a CST could indicate the existence of certain type of AMR gene or mutation. This seems not fitting well with the previous point of view based on AMR phenotype (21). A limitation of this study is that PCR specific for *S*. Pullorum wasn't carried out during the sampling periods to prove the dominance of *S*. Pullorum in *Salmonella* contamination in the breeding farms.

Collectively, this study demonstrates that CRISPR-MVLST is an ideal choice, close to CoreSNP based on WGS, for typing *Salmonella* as it well-distinguished the isolates in the same serovar or ST or even cgST. The study also shows that its advantage in tracing back to the contaminating isolates in the slaughterhouse and market. CRISPR-MVLST meets the requirements of large-scale epidemiological investigation and tracing the major lineages of *Salmonella* in the poultry production chain. Further studies, for samplings from different food commodities, a diverse *Salmonella* serovars, and isolates from a large time scale, are needed, as well as samplings in other scenarios such as outbreak investigation.

## MATERIALS AND METHODS

**Ethical approval.** The experimental protocols regarding the animal handlings were approved by the Laboratory Animal Management Committee of Zhejiang University (Approval No. 2015016).

**Sampling and isolation.** A total of 1,064 samples were collected from the poultry production chain, including breeding farms, slaughterhouses, and markets during 2015–2016 in Zhejiang province, China. The samples consisted of swabs ($n$ = 146) from breeding farms; water samples ($n$ = 153) and swabs of various sources ($n$ = 600) from slaughterhouses; and carcass swabs ($n$ = 165) from markets (Table S5). All samples were transported to the laboratory in ice packs on the same day. The swab samples were collected in 2 mL sterile phosphate-buffered saline (PBS) in Eppendorf (EP) tubes. The water samples (5 mL) from the disinfection tank and chilling tank in slaughterhouses were collected in sterile 7 mL Falcon tubes. For preliminary enrichment, Buffered Peptone Water (BPW, Haibo Biotechnology Co) was used in a 1:9 dilution (sample in PBS: BPW) and incubated at 37℃ for 16–18 h in a rotatory incubator set at 180 rpm. For selective enrichment, Tetrathionate Broth Base (TTB, Land bridge Biotechnology Co), supplemented with iodine solution (Land bridge Biotechnology Co) and brilliant green solution (Land bridge Biotechnology Co) was used at a dilution of 1:10 (sample in BPW: TTB) and incubated at 42℃ for 22–26 h in a rotatory incubator set at 180 rpm. For primary screening of *Salmonella*, bacterial DNA was extracted using the TIANamp bacteria DNA kit (Tiangen Biotech) according to the manufacturer's instructions, and the DNA was subjected to PCR based identification as described previously (47). Pure *Salmonella* colonies were isolated from the positive samples by subculturing the selectively enriched samples on Xylose Lysine Deoxycholate (XLD, Land bridge Technology Co) agar with an incubation of 18–22 h at 37℃. Typical and pure colonies were picked up after subculturing on XLD agar and were transferred intoLB broth and incubated for 18–22 h at 37℃ in a rotatory incubator set at 180 rpm. For confirmation of *Salmonella* isolates, genomic DNA was extracted using the TIANamp bacteria DNA kit from overnight cultures of the pure colonies and PCR was conducted using genus-specific primers as mentioned previously (47, 48).

**Serotyping.** The PCR confirmed *Salmonella* isolates were serotyped by the slide agglutination method as previously described (24, 49).

**Antimicrobial susceptibility.** The antimicrobial susceptibility of the confirmed isolates was performed by MIC assay using a panel of 15 antimicrobial agents, including penicillin (ampicillin: AMP, 0.25–128 $\mu$g/mL); $\beta$-lactamase inhibitor combinations (amoxicillin-clavulanic acid: AMC, 0.125/0.062–128/64 $\mu$g/mL); cephems (ceftiofur: CF, 0.125–128 $\mu$g/mL; cefoxitin: CX, 0.125–128 $\mu$g/mL); carbapenems (imipenem: IPM, 0.03–16 $\mu$g/mL), aminoglycosides (gentamicin: GEN, 0.031–64 $\mu$g/mL; kanamycin: KAN, 0.25–128 $\mu$g/mL; streptomycin: STR, 1–128 $\mu$g/mL); tetracyclines (tetracycline: TET, 0.062–128 $\mu$g/mL); (fluoro)quinolones (ciprofloxacin: CIP, 0.015–16 $\mu$g/mL; nalidixic acid: NAL, 0.5–128 $\mu$g/mL); sulfonamides (trimethoprim-sulfamethoxazole: TST, 0.25/4.75-32/608 $\mu$g/mL); polypeptides (colistin: COL, 0.031–64 $\mu$g/mL); macrolides (azithromycin: AZI, 0.25–128 $\mu$g/mL), and phenicol (chloramphenicol: CHL, 0.5–128 $\mu$g/mL), as elaborated in our previous studies (50–52).

**Multilocus sequence typing.** Genomic DNA of isolates was extracted as mentioned above and quantified using the Qubit Broad Range assay kit (Invitrogen, Carlsbad), as per the manufacturer's instructions. The 7-gene legacy MLST was conducted with the Sanger sequencing platform at Sunya Biotechnology Co., Ltd (Zhejiang, China) using the methods previously reported (53). Determination of STs was based on the sequence analysis of seven housekeeping genes using the Enterobase database (54). Among all confirmed isolates, 61 unique isolates were selected for further analysis after removing copy isolates in the same sample based on serovar, sequence type (ST) and colony morphology.

**Genome sequencing and assembling.** Whole-genome sequencing was performed on an Illumina Nextseq platform using PE150 strategies. The raw reads were checked for sequence quality as described previously (55). The quality of sequencing reads was checked by FastQC toolkit v0.72, and low-quality sequences or joint sequences were removed using Trimmomatic v0.39 as described (56). De novo assembly was performed using SPAdes v3.12.0 on an in-house Galaxy platform (31).

**CRISPR typing, CRISPR-MVLST, and data visualization.** To obtain more complete CRISPR sequences, the CRISPR1 and CRISPR2 sequences were collected using the Sanger sequencing platform of Sunya Biotechnology Co., Ltd (Zhejiang, China) according to the method described previously (19, 25–27). These sequences were consistent with 2 CRISPR loci's sequences extracted from WGS. Analysis of CRISPR1 and CRISPR2 was carried out by CRISPRCasFinder (57) to locate and obtain the spacer information contained in the corresponding isolates. A local database of CRISPR1 and CRISPR2 spacers was established by R v3.6.1 to project the binary matrix of CRISPR-pattern (combination of both CRISPR loci's spacers), which could be used for the phylogenetic tree, heatmap (produced by Microsoft Excel 2016), CT and CRISPR-MVLST. By GrapeTree v1.5.0, a neighbor-joining method (FastME V2) was used to calculate the tree (58). Simultaneously, a non-redundant database of CRISPR-pattern was built, and each CRISPR-pattern was numbered for CT.

Genome data (introduced here as a more efficient way, but Sanger sequencing data also works in the same manner) was imported into the in-house Galaxy platform as mentioned previously (59–63), and the location information of the corresponding *fimH* and *sseL* sequences of the isolates were detected by ABRicate v0.8 and Virulence Factor Database (VFDB) deployed to the platform (similarity >90%, coverage >60%) (64). Then, FASTA sequences were incorporated into Geneious v8.1.6 for visualization and extraction of these sequences. Compared with the previous literature (18, 26, 27), we numbered *fimH* and *sseL* sequences, assigned new numbers to the newly detected *sseL* and *fimH* sequences and build a non-redundant database.

Numbers of the above 4 loci (CRISPR1, CRISPR2, *fimH* and *sseL*) of each isolate were combined to produce the CRISPR-MVLST profiles (18, 26, 27), and a local CRISPR-MVLST sequence type (CST) database was established using a matrix consisting of these profiles. Finally, the CST number of each isolate was assigned using this database.

**Core genome multilocus sequence typing.** cgMLST analysis was performed using fastq data of all 61 isolates based on cgMLSTFinder software (65). An in-house Python3 script was used to convert the new cgMLST profile database matrix from Enterobase (http://enterobase.warwick.ac.uk/species/index/senterica, downloaded on 2021-04-23) to a usable format for cgMLSTFinder to produce updated cgMLST results.

**Whole-genome MLST.** Cano-wgMLST_BacCompare Platform was used (66) for wgMLST analysis. This platform employed 2 main processes, namely, whole-genome scheme extraction (GSE) and discriminatory loci refinement (DLR), and the "feature importance" algorithm was used (67). In the GSE step, "contig annotation" and "pan-genome allele database (PGAdb) creation" were used to process all 61 fasta format files and generate locus starting with "SAL". Finally, we obtained all strains' wgMLST allele_ profile matrix, and the number of types was calculated and assigned using this matrix following the similar method of CRISPR-MVLST.

**WGS-based CoreSNP typing.** *S.* Typhimurium LT2 was used as a reference genome, and 115,469 core genome single nucleotide polymorphisms (Core SNPs) were identified by Snippy v4.4.4 as described in our previous publications (5, 50, 55). Core SNPs alignment was transformed into profile matrix, which was used for CoreSNP typing following the similar method of CRISPR-MVLST.

**WGS-based plasmid profile typing.** The assembled genomes were analyzed for plasmid replicons based on the CGE PlasmidFinder database (similarity >95%, coverage >60%) (68) using ABRicate v0.8 (69) and in-house script as previously used (70). The produced 0/1 plasmid profile matrix (0 for absence, 1 for presence) was analyzed, following the similar method of CRISPR-MVLST.

**WGS-based AMR gene profile typing.** The assembled genomes were analyzed for AMR gene based on the CGE ResFinder database (68) (similarity >90%, coverage >60%) using ABRicate v0.8 deployed to the in-house Galaxy platform (69) and in-house script as previously used (70). The resulted 0/1 profile matrix (0 for absence, 1 for presence) was analyzed following the similar method of CRISPR-MVLST.

**WGS-based VF gene profile typing.** The assembled genomes were analyzed for potential virulence factor (VF) genes based on the Virulence Factor Database (VFDB) database (64) (similarity >90%, coverage >60%) using ABRicate v0.8 (69) and in-house script as previously used (70). The produced 0/1 profile matrix (0 for absence, 1 for presence) was analyzed following the similar method of CRISPR-MVLST.

**AMR phenotype-based typing.** For AMR profile typing method and MIC profile typing method, the number of types was calculated and assigned using corresponding profile matrix following the similar method of CRISPR-MVLST.

**The discriminatory power of the methods.** To evaluate the typing potential of the 12 studied methods, the individual Discriminatory Index (DI) was calculated based on the following formula (71):

$$DI = 1 - \sum_{i=1}^{S} Mi(Mi - 1)/N(N - 1)$$

where $N$ is the total number of strains in the sample population, $S$ is the total number of types described, and $Mi$ is the number of strains belonging to the i[th] type.

**Bioinformatics analysis for genomic epidemiology.** Serovar prediction for 61 studied isolates was carried out with 2 different methods, SISTR (72) and SeqSero2 (55, 73). These methods cross-validated the accuracy of prediction results. The Core SNPs alignment mentioned above was used for building a maximum-likelihood phylogenetic tree (1000 bootstraps) using IQ-TREE v1.6.12 (74) with the best model TVM+F+ASC+R3. The plasmid replicon, AMR gene, and VF gene were predicted or detected as mentioned above. The AMR mutations were detected by RGI v5.1.1 with CARD database v3.1.0 (similarity >90%, coverage >60%) as previously reported (70, 75). The plasmid, AMR determinant and VF gene with the phylogenetic tree were visualized by R-studio v1.1 with R v3.6.1 and R packages (ggplot2, ggtree, treeio, phytools, ape, maps, phangorn, Rcpp, vctrs, tidyverse, and gheatmap).

**Data availability.** All the raw data have been deposited into CNGB Sequence Archive (CNSA) (76) of China National GenBank DataBase (CNGBdb) (77) with a project title 'poultry production chain' and accession number CNP0001590.

## SUPPLEMENTAL MATERIAL

Supplemental material is available online only.
**SUPPLEMENTAL FILE 1**, PDF file, 0.7 MB.
**SUPPLEMENTAL FILE 2**, XLSX file, 3.2 MB.

## ACKNOWLEDGMENTS

This work was supported by the National Program on Key Research Project of China (2019YFE0103900) as well as the European Union's Horizon 2020 Research and Innovation Program under Grant Agreement No. 861917 – SAFFI, the National Program on Key Research Project of China (2018YFD0701001), the National Natural Science Foundation of China (31872837 & 32150410374), Natural Science Foundation of Zhejiang Province (LR19C180001), Zhejiang Provincial Key R&D Program of China (2022C02024, 2021C02008, 2020C02032), and China Postdoctoral Science Foundation (2022M712785).

We appreciate the efforts of all the participants in China who contributed to the sampling and data collection. We also thank all those individuals involved in the characterization of *Salmonella* isolates and giving advice, especially associate chief technician Xuebin Xu of Shanghai Municipal CDC, Miss Qingqing Wu, Xiao Zhou and Tanveer Muhammad of Zhejiang University.

M.Y., X. Li, and W.F. designed the study. H.P., C.J., N.P., and C.K. wrote the initial draft. H.P., C.J., F.L., and J.M. did genomic data analysis, figures, and tables. H.P., N.P., X. Liu, X. Liao, and X. Li collected the samples. N.P., F.L., J.M., X. Liu, C.D., K.Z., X. Liao, J.G., did the microbiological investigations. M.Y. conceived the idea, finalized the manuscript, and managed the project. All the authors contributed to the article and approved the submitted version.

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
