## [Reviewer comments · Microbiology Spectrum]

Microbiology Spectrum

Comprehensive Assessment of Subtyping Methods for Improved Surveillance of Foodborne Salmonella

Hang Pan, Chenghao Jia, Narayan Paudyal, Fang Li, Junyong Mao, Xi Liu, Chenghang Dong, Kun Zhou, Xiayi Liao, Jiansen Gong, Weihuan Fang, Xiaoliang Li, Corinna Kehrenberg, and Min Yue

Corresponding Author(s): Min Yue, Zhejiang University

Review Timeline:

Submission Date:	June 30, 2022
Editorial Decision:	August 5, 2022
Revision Received:	September 2, 2022
Accepted:	September 13, 2022

Editor: Xianqin Yang

Reviewer(s): The reviewers have opted to remain anonymous.

Transaction Report:

DOI: <https://doi.org/10.1128/spectrum.02479-22>

August 5, 2022

Prof. Min Yue
Zhejiang University
College of Animal Sciences
College of Animal Sciences
Hangzhou
China

Re: Spectrum02479-22 (Comprehensive Assessment in Subtyping Methods for Improved Implementation of Surveillance of Foodborne Salmonella)

Dear Prof. Min Yue:

Thank you for submitting your manuscript to Microbiology Spectrum. The work deals with a very important topic. However, much of the credit of the science is lost as it was very difficult to read/understand the paper. I am looking forward to seeing a more-focused and organized revision. When submitting the revised version of your paper, please provide (1) point-by-point responses to the issues raised by the reviewers as file type "Response to Reviewers," not in your cover letter, and (2) a PDF file that indicates the changes from the original submission (by highlighting or underlining the changes) as file type "Marked Up Manuscript - For Review Only". Please use this link to submit your revised manuscript - we strongly recommend that you submit your paper within the next 60 days or reach out to me. Detailed instructions on submitting your revised paper are below.

Link Not Available

Sincerely,

Xianqin Yang

Journals Department
Reviewer comments:

Reviewer #1 (Comments for the Author):

This paper compares 12 typing methods for Salmonella in the context of surveillance. Methods have a range of technical requirements and discriminatory properties, and the authors provide suggestions for best methods to use given a lab's capacity. Overall this paper deals with an important topic and contains useful information. However, it is very difficult to read and understand. A lot of data is presented and at times the focus of the paper is lost. Better organization of the paper might help. For example, several typing methods are lumped in the section "Genomic analysis of Salmonella isolates". Also it was difficult to tease out the methods used for all 12 typing schemes. It would be easier to follow if each typing method had its own section in the methods and in the results but I will leave this up to the authors. Figure S4 was especially helpful and could be in the main

manuscript.

Improved is spelled wrong in the title. Suggest rewording it to Comprehensive assessment of subtyping methods for improved surveillance of foodborne Salmonella

L66 - not sure what a non-copy strain is - do you mean unique?

L98 - this is an unusual start for a paper. I am wondering if the first few lines of the Introduction got cut-off.

L103 - provide reference for how antibiogram typing is used in the clinic

L233- 235. This can go in a figure legend

L236 - in house Galaxy? Provide details

L244 - state the 4 loci

General comment to include in Introduction: Are the number of CRISPR spacers consistent over time in Salmonella?

Discussion points to consider:

- Subtyping has many purposes - the best scheme for surveillance may not be the one to use for source attribution or outbreak investigations
- Consider stability of the method - plasmid profiling, MIC, antibiogram results might change more rapidly over time compared to other typing methods
- How would these results translate across commodities, serovars, time?

Reviewer #2 (Comments for the Author):

This reviewer appreciated the hard work of the authors in addressing my questions. One additional question here:

In this work, 61 isolates were used. Based on the gold standard typing method, 19 types were indicated. However, 30 types were detected by the use of your approach.

I am convinced that your approach showed a higher differentiation index than the gold standard approach. One important and fundamental question here is: Is the method detecting more types better than the one detecting less number of types? Would a method that detects 61 types from 61 isolates be the best typing method? Please discuss and provide the rationale and justification.

Staff Comments:

Preparing Revision Guidelines

Please return the manuscript within 60 days; if you cannot complete the modification within this time period, please contact me. If you do not wish to modify the manuscript and prefer to submit it to another journal, please notify me of your decision immediately so that the manuscript may be formally withdrawn from consideration by Microbiology Spectrum.

Corresponding authors may join or renew ASM membership to obtain discounts on publication fees. Need to upgrade your

membership level? Please contact Customer Service at Service@asmusa.org.

Dear Dr. Xianqin Yang,

We made the careful revisions and submit the improved manuscript for your consideration for the publication in ***Microbiology Spectrum***.

Re: Spectrum02479-22 (Comprehensive Assessment in Subtyping Methods for Improved Implementation of Surveillance of Foodborne Salmonella)

Dear Prof. Min Yue:

*Thank you for submitting your manuscript to Microbiology Spectrum. The work deals with a very important topic. However, much of the the credit of the science is lost as it was very difficult to read/understand the paper. I am looking forward to seeing a more-focused and organized revision. When submitting the revised version of your paper, please provide (1) point-by-point responses to the issues raised by the reviewers as file type "**Response to Reviewers**," not in your cover letter, and (2) a PDF file that indicates the changes from the original submission (by highlighting or underlining the changes) as file type "**Marked Up Manuscript - For Review Only**". Please use this link to submit your revised manuscript - we strongly recommend that you submit your paper within the next 60 days or reach out to me. Detailed instructions on submitting your revised paper are below.*

The ASM Journals program strives for constant improvement in our submission and publication process. Please tell us how we can improve your experience by taking this quick Author Survey.

Sincerely,

Xianqin Yang

We have made thorough changes and clarifications in the new manuscript, and accordingly, point-by-point response in the following. For the ease of track, we used the three colours in the text, with yellow for the editor, purple for reviewer 1, green for reviewer 2. We have also corrected other minor mistakes or polished the English and marked them up in the manuscript with cyan.

Editor comments:

The work deals with a very important topic. However, much of the the credit of the science is lost as it was very difficult to read/understand the paper. I am looking forward to seeing a more-focused and organized revision.

Thank you so much for your time and effort for improving our work. According to your important comments, we have made corresponding revisions of the text in yellow. We have revised the manuscript according to all your constructive suggestions, and tried our best to improve our manuscript (in the new Line 54-56, 148-342, 466~468, Table S5, 494-593, 875-876).

Reviewer comments:

Reviewer #1 (Comments for the Author):

This paper compares 12 typing methods for Salmonella in the context of surveillance. Methods have a range of technical requirements and discriminatory properties, and the authors provide suggestions for best methods to use given a lab's capacity. Overall this paper deals with an important topic and contains useful information. However, it is very difficult to read and understand. A lot of data is presented and at times the focus of the paper is lost. Better organization of the paper might help. For example, several typing methods are lumped in the section "Genomic analysis of Salmonella isolates". Also it was difficult to tease out the methods used for all 12 typing schemes. It would be easier to follow if each typing method had its own section in the methods and in the results but I will leave this up to the authors. Figure S4 was especially helpful and could be in the main manuscript.

Thank you so much for summarizing our work. According to your valuable comments and important suggestions, we have made corresponding revisions **of the text in purple.**

We have revised the manuscript according to all your kind comments, and tried our best to make Methods and Results parts easier to read and understand (in the **new Line, 168-342, 443-445, 494-593**, for reading ease, in Methods and Results parts only new title of each section, newly added text or significantly organized text were coloured). The contents of Results part were particularly rearranged according to the structure of Table 2, 3 for a better readability.

We set the old Fig. 3 as new Fig. S1. We also have moved the old Fig. S4 (now as **new Fig. 3**) into the main manuscript as you suggested (in the **new Line 247**). The other figures were renumbered accordingly.

Comments:

1) Improved is spelled wrong in the title. Suggest rewording it to Comprehensive assessment of subtyping methods for improved surveillance of foodborne Salmonella

Thank you so much for pointing this out, we have corrected it and reworded the title accordingly **in the new Line 1-3.**

2) L66 - not sure what a non-copy strain is - do you mean unique?

Thank you for pointing this out. We mean a unique isolate here and we have changed the word in the **new Line 46, 165 and 514**.

3) L98 - *this is an unusual start for a paper. I am wondering if the first few lines of the Introduction got cut-off.*

Thank you for pointing this key point out, and we have modified accordingly and smooth the start in the **new Line 81-90**.

4) L103 - *provide reference for how antibiogram typing is used in the clinic*

Thank you for your critical comments. To our knowledge, very few clinic labs conducted antibiogram typing for such purpose. However, the valuable parameters, particularly the MICs, can be used as the important classifier for different strains. Our previously paper has already demonstrated the power by only of MICs value can guide the understanding of *Salmonella* from different sources (*Frontiers in Microbiology* 2018, 9:23. DOI: 10.3389/fmicb.2018.00023.) and via time (*Frontiers in Microbiology* 2019, 10:985. DOI: 10.3389/fmicb.2019.00985.), and different feeding styles (*Journal of Hazardous Materials* 2022, 438:129476. DOI:10.1016/j.jhazmat.2022.129476). All together, we believe the antibiogram typing remains the important choice for epidemiological surveillance purpose. We have modified the text and added all corresponding references accordingly **in the new Line 90-92**.

5) L233- 235. *This can go in a figure legend*

Thank you for your valuable suggestions. We have modified the text accordingly **in new Fig 2's legend at the end of the manuscript**.

6) L236 - *in house Galaxy? Provide details*

Thank you for your suggestion. Actually it means the “in-house Galaxy platform”, we mentioned it as a local Galaxy platform based on the core knowledge from literature (***Nucleic Acids Res.* 2016, 44(W1):W3-W10. doi: 10.1093/nar/gkw343**) with abricate software and VFDB database installed for detecting VF genes and locating the sequences of *fimH* and *sseL*.

Furthermore, the antimicrobial resistance genes (ARG) and plasmid types were also detected using assemblies on the in-house Galaxy platform as previously reported in our recent publications (*Frontiers in Public Health* 2022, DOI: 10.3389/fpubh.2022.988317; *Critical Reviews in Food Science and Nutrition* 2022, 1-19. DOI: 10.1080/10408398.2022.2087174; *Antibiotics* 2022, 11(5), 625. DOI:10.3390/antibiotics11050625; *Microbiology Spectrum* 2022, e0096522. DOI:10.1128/spectrum.00965-22; *International Journal of Food Microbiology* 2022, 366: 109572. DOI: 10.1016/j.ijfoodmicro.2022.109572; *Frontiers in Medicine* 2022, 9:753085. DOI: 10.3389/fmed.2022.753085). We also improved the text accordingly in new Line 538-541.

7) L244 - state the 4 loci

Thank you for your kind suggestion. We have added the context accordingly in new Line 546.

8) General comment to include in Introduction: Are the number of CRISPR spacers consistent over time in *Salmonella*?

Thank you for your suggestion. Actually, we have found the No. of CRISPR spacers is less consistent over time in *Salmonella* and probably will add/delete more spacer after shifting to another circumstance. This adds the diversity in *Salmonella* CRISPR-pattern or CRISPR loci' array, usually reflecting *Salmonella* strain's geographical location shift. And that's why it is an attractive locus to be included for typing purpose.

We have modified text accordingly in new Line 125-137.

Discussion points to consider:

1) Subtyping has many purposes - the best scheme for surveillance may not be the one to use for source attribution or outbreak investigations

Thank you for giving such important point suggestion. Routine epidemiological surveillance means regular or certain network-based surveillance. **Identifying every lineage (based on adequate discriminatory power)** may be the most important for typing method during an epidemiological surveillance investigation.

For source attribution, frequency-matching models are often used. They rely on the one-

to-one matching of microbial subtypes in humans and sources, and have been extensively used for source attribution of major (bacterial) foodborne pathogens. If no clear or **representative genetic relationship among all isolates** is identified by a typing method, it will be much confusing and the source attribution or contamination tracking may be misled. For example, CRISPR-MVLST showed much greater utility in tracking major lineages and ecological sources than MLVA (*J Clin Microbiol.* 2015 Jan;53(1):212-8. doi: 10.1128/JCM.02332-14).

A method for an outbreak investigation usually needs to be **good and accurate in discerning the genetic relationship of two isolates**, and traditional microbiological methods are not suited to attribute pathogens due to a low genotyping/phenotyping diversity.

For better clarification this point, we have modified the text accordingly **in the new Line 36-37, 365-367 and 455-456.**

2) Consider stability of the method - plasmid profiling, MIC, antibiogram results might change more rapidly over time compared to other typing methods

Thank you for your suggestion. We agree with your point. The stability of the method will be poor because these methods are not standardized in choosing loci, and may bring uncertain loci in any case. That's why we don't recommend to convey these typing in all cases. We choose these methods for only comparative purpose. We have added this important point in the discussion accordingly **in the new Line 405-407.**

3) How would these results translate across commodities, serovars, time?

Thank you for your insightful comments. We believe the provided options, an array of 12 typing methods in this study, could guide future study by focusing on important issue such as food commodities, serovars and time. Additionally, we have added this important point in the discussion **in the new Line 453-455.**

Reviewer #2 (Comments for the Author):

This reviewer appreciated the hard work of the authors in addressing my questions. One additional question here: In this work, 61 isolates were used. Based on the gold standard typing method, 19 types were indicated. However, 30 types were detected by the use of your approach. I am convinced that your approach showed a higher differentiation index than the gold standard approach.

Thank you so much for summarizing the work and giving suggestions. We really appreciate your precious time and great effort in improving our manuscript. According to your important question, we have made corresponding revisions of the text in green.

- One important and fundamental question here is:

Is the method detecting more types better than the one detecting less number of types? Would a method that detects 61 types from 61 isolates be the best typing method? Please discuss and provide the rationale and justification.

Thank you for pointing this out. We have discussed and provided the rationale and justification as far as we learned here:

Firstly, **the number of types only reflects the range of existing differences**, while the method with a result of less types can have a higher DI than the method with a result of more types, such as CRISPR-MVLST which presents less types but higher DI compared with AMR gene profile and VF gene profile method, as we suggested in the Table 3. Even if we found a method showing its DI=1.0, we still cannot determine whether this method can be truly applied in certain scenarios.

Secondly, because the typing method can be used **in many scenarios**, such as routine epidemiological surveillance, outbreak investigation, source attribution or risk assessment, which usually requires **different levels of resolution or ease of use for the different purposes**.

Take clustering/source attribution/tracking as an example, if we identify 61 types from 61 isolates but find **no clear or representative genetic relationship of these isolates**, it will be much confusing and may mislead the **clustering, source attribution, contamination tracking**. For example, compared with MLVA, CRISPR-MVLST showed much greater utility in tracking major lineages and ecological sources (*J Clin Microbiol.* 2015 Jan;53(1):212-8. doi: 10.1128/JCM.02332-14).

Considering an ease of use for such technique, MLVA is more appropriate for small scale

of investigation. In contrast, CRISPR-MVLST can deliver more provisional sequence data, for large-scale analysis such as cross-laboratories, -regions, -counties, but this method usually needs more effort for extracting the sequencing data, in particularly from the CRISPR regions, which is not convenient to access. Additionally, if we cannot get a representative information of phylogenetic relationship during surveillance studies, better strategies, including whole genomic sequencing, are also necessary.

Thirdly, if a method misses or **cannot discern the genetic relationship of two isolates**, it probably presents a **reduced epidemiological concordance** even if its DI is the highest. For this reason, some researchers proposed that CRISPR-MVLST can be used as an alternative to PFGE (even with the highest DI) in some serovars (*BMC Microbiol* 2013, 13, 254. doi:10.1186/1471-2180-13-254). They provided a good example: a S. Heidelberg outbreak associated with ground turkey in 2011 involved isolates (all genetically close) with two distinctly different PFGE patterns, indicating that PFGE could not discern the close genetic relationship between some isolates (Investigation Update: Multistate Outbreak of Human Salmonella Heidelberg Infections Linked to Ground Turkey. <https://www.cdc.gov/salmonella/heidelberg/111011/timeline.html>).

Fourthly, all parameters including repeatability, level of throughput, cost and time should to be considered before the implementation of the surveillance investigation.

We added your valuable points in our discussion, and further clarified our concept and opinion, and made corresponding improvements related to this discussion in **new Line 65-66, 224-225 and 455-456**.

Again, we thank the editor and two anonymous reviewers for their time and effort in improving our manuscript. We believe that our revision is satisfactory. If you require any additional information regarding our manuscript, please do not hesitate to contact us.

Stay safe and well,

Professor of Microbiology

September 7, 2022

Prof. Min Yue
Zhejiang University
College of Animal Sciences
College of Animal Sciences
Hangzhou
China

Re: Spectrum02479-22R1 (Comprehensive Assessment of Subtyping Methods for Improved Surveillance of Foodborne Salmonella)

Dear Prof. Min Yue:

Your manuscript has been accepted, and I am forwarding it to the ASM Journals Department for publication. You will be notified when your proofs are ready to be viewed.

Sincerely,

Xianqin Yang
Editor, Microbiology Spectrum

Journals Department
Supplemental Material FOR Publication: Accept
Supplemental Material FOR Publication: Accept